# A Retrospective Multicenter Italian Analysis of Epidemiological, Clinical and Histopathological Features in a Sample of Patients with Acinic Cell Carcinoma of the Parotid Gland

**DOI:** 10.3390/cancers15225456

**Published:** 2023-11-17

**Authors:** Pietro De Luca, Arianna Di Stadio, Luca de Campora, Egidio De Bonis, Matteo Fermi, Gerardo Petruzzi, Francesca Atturo, Roberta Colangeli, Alfonso Scarpa, Alfredo Lo Manto, Andrea Colizza, Giulia Cintoli, Giulia Togo, Giovanni Salzano, Domenico Crescenzi, Massimo Ralli, Vincenzo Abbate, Filippo Ricciardiello, Luciano Magaldi, Aurelio D’Ecclesia, Gianluca di Massa, Leopoldo Costarelli, Elisabetta Merenda, Alessandro Corsi, Renato Covello, Rosa Maria Di Crescenzo, Loren Duda, Lucia Maria Dimitri, Alessandro Caputo, Gerardo Ferrara, Teresina Lucante, Francesco Longo, Domenico Tassone, Maurizio Iemma, Michele Cassano, Francesco Antonio Salzano, Luigi Califano, Daniele Marchioni, Raul Pellini, Marco de Vincentiis, Livio Presutti, Franco Ionna, Enrico de Campora, Marco Radici, Angelo Camaioni

**Affiliations:** 1Otolaryngology Department, Fatebenefratelli Isola Tiberina-Gemelli Isola, 00186 Rome, Italy; domenico.crescenzi@fbf-isola.it (D.C.); radix997@hotmail.it (M.R.); 2Otolaryngology Unit, University of Catania, 95131 Catania, Italy; 3Otolaryngology Department, San Giovanni-Addolorata Hospital, 00100 Rome, Italyfatturo@hsangiovanni.roma.it (F.A.); datassone@hsangiovanni.roma.it (D.T.); acamaioni@hsangiovanni.roma.it (A.C.); 4Otolaryngology Unit, San Giovanni di Dio e Ruggi D’Aragona Hospital, 84131 Salerno, Italy; egidio.debonis@sangiovannieruggi.com (E.D.B.); maurizio.iemma@sangiovannieruggi.it (M.I.); 5Department of Otorhinolaryngology—Head and Neck Surgery, IRCCS Azienda Ospedaliero-Universitaria di Bologna, 40138 Bologna, Italy; matteo.fermi.med@gmail.com (M.F.); livio.presutti@unibo.it (L.P.); 6Department Otolaryngology Head and Neck Surgery, IRCCS Regina Elena National Cancer Institute, Istituti Fisioterapici Ospitalieri (IFO), 00144 Rome, Italy; gerardo.petruzzi@ifo.it (G.P.); raul.pellini@ifo.it (R.P.); 7Otolaryngology Department, Sant’Eugenio Hospital, 00144 Rome, Italy; roberta.colangeli@aslroma2.it; 8Department of Medicine, Surgery and Dentistry, University of Salerno, 84081 Salerno, Italy; alfonso.scarpa@sangiovannieruggi.com (A.S.); fasalzano@unisa.it (F.A.S.); 9Otolaryngology Department, University of Modena and Reggio Emilia, 41121 Modena, Italy; alfredo.lomanto@unibo.it (A.L.M.); daniele.marchioni@unimore.it (D.M.); 10Department of Sense Organs, University Sapienza, 00161 Rome, Italy; andrea.colizza@uniroma1.it (A.C.); massimo.ralli@uniroma1.it (M.R.); marco.devincentiis@uniroma1.it (M.d.V.); 11Otolaryngology Unit, University of Foggia, 71122 Foggia, Italy; giulia.ciintoli@unifg.it (G.C.); luciano.magaldi@unifg.it (L.M.); michele.cassano@unifg.it (M.C.); 12Maxillofacial Surgery Unit, Department of Neurosciences, Reproductive and Odontostomatological Sciences, University of Naples “Federico II”, 80100 Naples, Italy; giulia.togo@unina.it (G.T.); giovanni.salzano@unina.it (G.S.); vincenzo.abbate@unina.it (V.A.); califano@unina.it (L.C.); 13Otolaryngology Unit, AORN Cardarelli, 80131 Naples, Italy; filippo.ricciardiello@aocardarelli.it; 14Maxillofacial and Otolaryngology Unit, IRCCS “Casa Sollievo della Sofferenza” San Giovanni Rotondo, 71013 Foggia, Italy; aurelio.decclesia@operapadrepio.it (A.D.); f.longo@operapadrepio.it (F.L.); 15Department of Medical and Surgical Sciences for Children and Adults, University of Modena and Reggio Emilia, 41124 Modena, Italy; gianluca.dimassa@unimore.it; 16Department of Pathology, San Giovanni Addolorata-Hospital, 00184 Rome, Italy; lcostarelli@hsangiovanni.roma.it; 17Department of Molecular Medicine, Sapienza University of Rome, 00185 Rome, Italy; elisabeha.merenda@uniroma1.it (E.M.); alessandro.corsi@uniroma1.it (A.C.); 18Department of Pathology, IRCCS-Regina Elena National Cancer Institute, 00144 Rome, Italy; covello@ifo.it; 19Institute of Experimental Endocrinology and Oncology (IEOS), National Research Council, 80131 Naples, Italy; rosamaria.dicrescenzo@unina.it; 20Pathology Unit, Department of Clinical and Experimental Medicine, University of Foggia, 71122 Foggia, Italy; loren.duda@ospedaliriunitifoggia.it; 21Department of Pathology, IRCCS “Casa Sollievo della Sofferenza” San Giovanni Rotondo, 71013 Foggia, Italy; lm.dimitri@operapadrepio.it; 22Pathology Unit, San Giovanni di Dio e Ruggi D’Aragona University Hospital, 84131 Salerno, Italy; acaputo@unisa.it; 23Department of Pathology, INT-IRCCS “Fondazione G. Pascale”, 80131 Naples, Italy; gennaro.ferrara@istitutotumori.na.i; 24Department of Pathology, Fatebenefratelli Isola Tiberina-Gemelli Isola, 00186 Rome, Italy; teresina.lucante@fbf-isola.it; 25Maxillofacial Unit, INT-IRCCS “Fondazione G. Pascale”, 80131 Naples, Italy; f.ionna@istitutotumori.na.it; 26Associazione Ospedaliera Italia Centro-Meridionale Otorinolaringoiatrica (AOICO), 00100 Rome, Italy; decampora@gmail.com

**Keywords:** parotid, acinic cell carcinoma, parotidectomy, neck dissection, salivary gland

## Abstract

**Simple Summary:**

Acinic cell carcinoma (AciCC) is a rare parotid gland tumour that is indolent in the majority of patients; however, its definition is still evolving because in a subgroup of patients it shows aggressive behaviour with lateral neck metastases and massive recurrences. Several authors have attempted to characterise this subgroup of patients, but the main limitation is the rarity of the tumour for which there are no prospective studies. In this study, we defined the epidemiological, clinical and histological features of 77 patients with AciCC of the parotid gland, with the aim of defining the characteristics of the high-risk patients.

**Abstract:**

Background. The acinic cell carcinoma (AciCC) of the parotid gland is a rare tumor with an indolent behavior; however, a subgroup of this tumor presents an aggressive behavior with a tendency to recur. The aim of this multicenter study was to identify and stratify those patients with AciCC at high risk of tumor recurrence. Methods. A retrospective study was carried out involving 77 patients treated with surgery between January 2000 and September 2022, in different Italian referral centers. Data about tumor characteristics and its recurrence were collected. The histological specimens and slides were independently reviewed by a senior pathologist coordinator (L.C.) and the institution’s local head and neck pathologist. Results. The patients’ age average was 53.6 years, with a female prevalence in the group. The mean follow-up was 67.4 months (1-258, SD 59.39). The five-year overall survival (OS) was 83.2%. The 5-year disease-free survival (DFS) was 60% (95% CI 58.2–61.7). A high incidence of necrosis, extraglandular spread, lymphovascular invasion (LVI), atypical mitosis, and cellular pleomorphism was observed in the high-risk tumors compared to the low-risk ones. Conclusion. AciCC generally had an indolent behavior, optimal OS, DFS with few cervical node metastases, and rare distant relapses. This multicenter retrospective case series provides evidence of the need for clinical–epidemiological–histological stratification for patients at risk of poor outcomes. Our results suggest that the correct definition of high-risk AciCC should include tumor size, the presence of necrosis, extraglandular spread, LVI, atypical mitosis, and cellular pleomorphism.

## 1. Introduction

Malignant salivary cancers are rare, only representing 3% of all head and neck cancers, and 80% of these cancers are benign epithelial tumors and arise in the parotid gland [1].

Acinic cell carcinoma (AciCC) is the third most common malignant tumor of the parotid gland (12% incidence of all salivary malignancies). The World Health Organization (WHO) classified this tumor as “malignant epithelial neoplasm of salivary glands in which at least some of the neoplastic cells demonstrate serous acinar cell differentiation, which is characterized by cytoplasmic zymogen secretory granules. Salivary ductal cells are also a component of this neoplasm” [2]. Nevertheless, debates surrounding this definition are still “ongoing”.

In 1953, Goodwin was the first to classify AciCC as a “benign adenoma” or “acinic cell tumor” [3]; later, Buxton et al. described the first case of AciCC with a malignant behavior [4].

In 1988, Stanley et al. identified a subgroup of AciCC tumors with high-grade (HG) transformation; the authors defined HG AciCC as “areas of dedifferentiated high-grade (HG) adenocarcinoma” in association with areas of low-grade AciCC [5]. HG AciCC was associated with a higher risk of adverse outcomes (local recurrence or distant metastasis) than traditional AciCC; this high-grade form showed disease-related mortality ranging from 33% to 75% [6,7,8].

HG-mutated tumors have a conventional low-grade component characterized by specific microscopic and immunohistochemical features for a given entity, intermingled with or juxtaposed to areas of HG morphology.

We have already shown that recurrence or metastasis, as well as age >65 years, are independent prognostic indicators of tumor aggressiveness [9]. In contrast to other authors [10,11], we showed that the HG phenotype could not be predictive of recurrence or more aggressive behavior. However, our previous study did not evaluate histological tumor features. To date, despite several attempts, there is still a lack of a reliable histological grading of the AciCC of the parotid gland.

This retrospective, multicenter study aimed to analyze the impact of demographic, clinical, surgical, and histological characteristics of the parotid gland AciCC on survival outcomes. Moreover, through patient stratification, we aimed to identify subjects at high risk of adverse outcomes. To the best of our knowledge, this is the first European study in which all prognostic histological factors of the AciCC of the parotid gland are analyzed.

## 2. Materials and Methods

### 2.1. Objectives

The primary endpoint of this study was assessing the survival outcomes (progression-free survival (PFS) from diagnosis to first progression and overall survival (OS)) of patients who underwent the removal of the AciCC of the parotid gland. Then, the secondary endpoints were (i) the evaluation of the presence of clinical prognostic factors and (ii) the description of the progression sites. Finally, the third objective was to evaluate the role of histological patterns on survival outcomes, seeking to elaborate a predictive model to identify patients at high risk of negative outcomes.

### 2.2. Patient Population

Patients with confirmed biopsy of the AciCC of the parotid gland who underwent surgery between January 2000 and September 2022 were retrospectively included in this study. The cohort included patients from multicenter sites, recruited in the departments of Otolaryngology—Head and Neck Surgery of different Italian tertiary referral centers.

We retrospectively reviewed patients’ medical records including the type of performed surgery, histopathologic results, and follow-up reports. For each patient, the following data were collected: gender, age at the time of the diagnosis, fine-needle aspiration biopsy (FNAB) results, the type of performed surgery (including eventual neck dissection at the time of first surgery), the presence of clinically evident lymph node (s) (cN) or distant metastasis at the time of diagnosis, positive or negative margins of the tumor resection post-surgery, the use (or not) of adjuvant radiochemotherapy (CHRT), the presence (or not) of locoregional or distant recurrences, and the health status at the follow-up (including the time of follow-up, months). TNM classification system (version 8) [12] was used to stage the tumor. The data concerning survival outcomes were extracted from mortality registries, outpatient visit notes, and radiological follow-ups.

Exclusion criteria were (i) the absence of follow-up; (ii) patients affected by secondary metastatic disease involving the parotid gland; (iii) changes in the histological diagnosis during the review of the histological slides; and (iv) histological slides not available for reviewing.

The data collected from each center were inserted in a shared Excel file, which, once completed with all data, was used to perform statistical analyses. The data from all centers were analyzed by the first two authors (P.D.L. and A.D.S.) and then shared with the other researchers to review the results and draw conclusions.

### 2.3. Pathology

Each center collected the histological slides. All centers used the same protocols for the fixation and pathological examination of the tissues. All the pathologists who analyzed the specimens had over 15 years of experience and were blinded to clinical information, regimens of treatment, and study endpoints. In addition, the senior pathologist coordinator was blinded to the interpretation of the slides performed by other pathologists.

All histological slides were reviewed independently by two pathologists: the senior pathologist coordinator (L.C.) and the institution’s local head and neck pathologist.

According to previous studies and the recent definition based on the WHO Classification of Head and Neck Tumors [13], specific histopathological aspects were considered, namely (i) the growth pattern of the tumor (solid, trabecular, cribriform, microcystic, papillary cystic, follicular, or more than one pattern); (ii) the grade of the tumor (low or high grade, according to the definition from Skalova et al.) [6]; (iii) the margin of resection post-surgery, according to Hermanek and Wittekind (R0, Rclose, R1, or R2) [14]; (iv) tumor necrosis (absent, microfocal < 1 mm, macrofocal > 1 mm, or diffuse); (v) perineural invasion (absent or present); (vi) lymphovascular invasion (LVI) (absent, focal or <2 figures of LVI, or diffuse or >2 figures of LVI); (vii) extraglandular growth (absent or present); (viii) cellular pleomorphism (absent, mild, moderate, or severe); (ix) lymphoid stroma (LS) (negative or <1%, mild or 1–10%, moderate or 11–50%, or severe or >50%); (x) atypical mitoses (absent or present); (xi) mitotic index (per 10 HPFS), according to the study by Xu et al. [15] (0–1, 2–4, >5); (xii) neuroendocrine differentiation (chromogranin A positivity) (absent or present); (xiii) immunohistochemistry and special stain techniques used (PAS, mucicarmine, S100, cytokeratin, DOG1, others, or more than one special techniques); (xiv) stromal hyalinization (absent or present); (xv) the percentage of expression of ki67, according to the suggestion by Vacchi-Suzzi et al. [16]; and (xvi) the percentage of tumor infiltrating lymphocytes, according to the study by Salgado et al. [17]. Microscopically, tumor grade and tumor stage were noted.

The special immunohistochemistry stain techniques used were PAS and PAS-D, p63, S100, and DOG1 (Figure 1).

### 2.4. Statistical Analysis

Statistical tests were used to analyze the data. A two-tailed *t*-test (τ) was used to compare numeric parameters, a chi-square (χ) test was used to analyze the nominal data, and Cohen’s d was used to evaluate the impact of the different sample sizes on the results. To better evaluate the findings with several patterns, i.e., tumor necrosis, the nominal data were changed to numerical ones by assigning a number to each characteristic.

Because we collected both clinical and histological data, multilinear regression analyses were performed using these data separately to better understand which of the two types (clinical or histological findings) could have a higher impact. At first, multilinear regression analyses were performed to analyze the effect of clinical parameters (sex, age, margin, T, N, neck dissection, and adjuvant therapy) on the months of survival; then, the same test was performed to analyze the effect of histological findings (size, necrosis, extraglandular, lymphovascular, atypical mitosis, neuroendocrine, and pleomorphism) on the months of survival. To define which findings to include in the multivariate analyses, we referred to the parameters that had statistically significant differences in the first part of the statistical analyses (*t*-test and chi-square).

The *p* value was considered statistically significant < 0.05. The analyses were performed using Stata^®^, version 17 (StataCorp LLC, College Station, TX, USA).

### 2.5. Ethical Considerations

Ethical approval was waived by the local ethics committee in view of the retrospective non-interventional nature of the study. This study was conducted in compliance with the Helsinki Declaration. All the collected data were recorded in a computerized database. Patients who were still alive at the time of the enrollment were informed about the study via telephone, and none expressed opposition to inclusion.

## 3. Results

### 3.1. Population

In total, 106 patients with a diagnosis of AciCC were identified during the timeframe of this study; of these, 18 were excluded because of the unavailability of histological slides, and 11 were excluded due to the lack of follow-up. The definitive study group included 77 (53 women and 24 men) patients with AciCC age 53.6 ± 18.2 yr (CI95%: 12–87). The peak of incidence was in the fifth decade (*n* = 18, 23.4%). However, most of the patients (*n* = 45, 58.4%) were in their forties, fifties, and sixties at the time of diagnosis. Only six patients (7.8%) had preoperative clinical evidence of cervical node involvement (cN+). Table 1 summarizes the details of patients’ characteristics.

Overall, 69 patients (89.6%) underwent preoperative FNAB, which was positive for AciCC in only 3 cases (18.8%). Notably, 100% agreement was observed between the results of the preoperative FNAB and the final diagnosis in the three patients who were positive for AciCC as a result of the FNAB. There was a concordance between FNAB results and final diagnosis in 18.8% of the cases. Of the 69 patients who underwent FNAB, 40 patients (58%) had negative test results (false-negative results; not diagnostic), and in 16 cases (23.2%), the patients were affected by other non-AciCC tumors, the most prevalent of which was pleomorphic adenoma (11 people, 68.7% of cases).

None of the patients at the time of the initial diagnosis of AciCC was affected by extranodal metastasis.

### 3.2. Treatment Characteristics

All patients were treated with surgery; the surgical procedure was chosen based on the tumor location and extension of the disease. According to the European Salivary Glands Society (ESGS) classification of parotidectomies [18], 39 patients (50.6%) underwent superficial parotidectomy (I–II), 33 (37.7%) underwent total parotidectomy (I–IV), 4 (5.2%) underwent superficial parotidectomy extended to the inferior deep lobe (I–II–III), 4 (5.2%) underwent the enucleation of the lesion, and 1 patient (1.3%) underwent total parotidectomy with sacrifice of facial nerve (I–IV[VII]).

Neck dissection (ND) was not considered in the surgical plan for 64 patients (82.1%), while in 13 cases (16.9%), it was considered. In addition, 8 of the 13 patients (61.5%) underwent selective ND (levels II–IV).

Among the 77 patients included in this study, 17 (22.1%) received adjuvant radiotherapy (RT); of these, 12 (70.6%) underwent locoregional irradiation even in the absence of cervical lymph node invasion. Among the thirteen patients for whom ND was performed, nine (69.2%) were treated with adjuvant RT, two patients (2.6%) underwent chemotherapy (CH), and one received concomitant CHRT (1.3%).

### 3.3. Staging

The definitive stage of the disease is illustrated in Table 1. Most of the patients did not present cervical nodal metastases. Seven patients (9.1%) had a postoperative positive N stage. Three cases were pN1 (42.8%), and four patients were pN+ (2 pN2a (28.6%) and 2 pN2b (28.6%)). None of the patients were pN2c (bilateral cervical nodal metastases) or pN3a (nodal metastases with the largest diameter > 60 mm).

### 3.4. Survival Outcomes

Follow-up data were available for all patients. The mean follow-up was 67.4 ± 59.4 months (CI95%: 1–258)]. The 5-year overall survival (OS) was 83.2% (Figure 2a). The 5-year disease-free survival (DFS) was 60% (95% CI 58.2–61.7) (Figure 2b).

At the end of the study (June 2023), 62 patients (80.5%) were still alive without any evidence of disease (NED); 2 (2.6%) were alive, after the recurrence of the condition, without any evidence of disease (NEDII); and 2 (2.6%) were alive with recurrence (AWD). Six patients (7.8%) suffered from disease recurrence, with a median interval to the first recurrence of 47.4 ± 80.6 months (CI95%: 4–227). Only one patient was affected by lateral skull extension at the time of the diagnosis, and nobody experienced distant recurrences. Eight patients (10.4%) died due to other causes (DOOC), and three (3.9%) died due to disease-related consequences (DOD). Survival data and univariate analysis of the most relevant prognostic factors are shown in Figure 1.

### 3.5. Histopathological Analysis

The histopathological features of the entire cohort of patients are summarized in Table 2 and Table 3.

The tumor was <5 cm in 68 patients (88.3%), and in 9 cases (11.7%), it was larger than 5.1 cm. Overall, 64 tumors (83%) were low-grade tumors, and 13 (17%) were HG tumors.

Notably, 26 tumors had a solid growth pattern (33.8%) 12 had a microcystic pattern (15.6%), and 9 had a follicular pattern (11.7%). In addition, 39 cancers (50.6%) presented multiple growth patterns, and 38 (49.4%) exhibited a single growth pattern.

In 54 cases, the tumor did not present necrosis (70.1%), and the ones who had necrosis showed a diffuse phenomenon. Overall, 64 (83.1%) had negative PNI, 56 (72.7%) had a negative LVI, and 49 (63.6%) did not present extraglandular growth. Furthermore, 31 (40%) cases showed pleomorphism, and 46 did not (60%). In 43 patients (55.8%), the tumor was positive for lymphoid stroma (44.2%), and in 67.87%, it was without atypical mitosis. Notably, 2 cases (2.6%) had neuroendocrine differentiation, 47 (61%) exhibited stromal hyalinization, and 50 (65%) had a poor mitotic index.

For 52 cases (67.5%), we did not have data about ki67 expression; of the 25 patients (32.5%) who had this info, 13 (16.9%) showed a significant expression (>15%) of ki67.

TIL evaluation was available for all the patients in this study; overall, 20 patients (26%) showed no TILs, while 57 (74%) showed TILs in the tumor tissue; of those, 13 (22.8%) showed a TIL expression of >20%.

### 3.6. Prognostic Factors for Survival

The comparison between the HG AciCC and low-grade cases, and between the group of patients who experienced recurrence and those without relapse is summarized in Table 4 and Table 5.

#### 3.6.1. Clinical Parameters

The stratification of the patients based on tumor grade (high versus low grade) revealed 13 subjects affected by HG tumor (16.8%). The remaining 64 patients (83.2%) suffered from low-grade tumors. Two patients (15.4%) in the HG group presented a negative outcome, while only three (4.6%) in the low-grade group had negative outcomes. No statistically significant differences were observed among the two groups (χ: *p* = 0.1).

Stratification for age showed higher age in the HG group (average 71.7 +10.5) than in the low-grade group (average 49.9 ± 17.4); this difference was statistically significant (τ: *p* = 0.000023). Cohen’s d was 0.8 (CI 95%: −1.949–−0.686) (large effect).

Stratification for sex showed 38.5% male prevalence (5 subjects) in the HG group and 26.6% male prevalence (17 patients) in the low-grade group. No statistically significant differences were observed regarding this finding (χ: *p* = 0.5).

Patients in the low-grade group had longer disease-free survival than the HG group, respectively, 71.9 ± 63.6 (CI95%: 1–258) and 45 ± 26.9 (CI95%:12–112). The difference was not statistically significant (τ: *p* = 0.06), but it is important to consider that Cohen’s d was 0.7 (CI 95%: −1.368–−0.151); the small sample size of the HG group could largely interfere with *p* value.

T and N staging scores were worse in the HG tumor than in the low-grade group; the differences were statistically significant, respectively, (τ) *p* = 0.000093 and *p* = 0.001. No statistically significant differences were observed in terms of metastasis incidence among the two groups (χ: *p* = 0.2).

The type of surgery used was not statistically different among the groups (τ: *p* = 0.4); however, the margins of resection were wider in HG tumors than in low-grade tumors, with a statistically significant value (τ: *p* = 0.003). Finally, neck dissection was more necessary in HG tumors than in the low-grade cases (τ: *p* = 0.001), and this was also the case in terms of the need for adjuvant therapy (τ: *p* = 0.02).

#### 3.6.2. Histological Characteristics of the Tumor

In HG AciCC cases, HG areas were identified between 5% and 15% of the total tissue area.

Stratification for tumor size showed statistically significant differences (τ: *p* = 0.000751) between HG tumors (45.6 mm ± 21.4) and low-grade tumors (25.8 mm ± 15.6). Cohen’s d was 0.8 (CI 95%: −1.813–−0.563) (large effect). The incidence of necrosis was higher in HG tumors than in low-grade tumors, with a statistically significant value (τ: *p* = 0.001). The extraglandular growth was statistically significantly different between high- and low-grade tumors (χ: *p* = 0.009), and it was observed with high prevalence in HG tumors; in HG tumors, the lymphovascular invasion was very common, and its difference with that in low-grade cases was statistically significant (τ: *p* = 0.006).

Similar results were observed in terms of the presence of atypical mitosis (high > low) (χ: *p* < 0.00001), neuroendocrine differentiation (more in the HG) (χ: *p* = 0.01), and pleomorphism (higher in high-grade tumors than in low-grade tumors) (χ: *p* = 0.003).

No statistically significant differences were observed comparing perineural invasion between high- and low-grade tumors (χ: *p* = 0.1); a statistically significant value was also not observed for stromal hyalinization (χ: *p* = 0.6), TILs (τ: *p* = 0.2), mitotic index (τ: *p* = 0.3), lymphoid stroma (τ: *p* = 0.1), the main growth pattern (τ: *p* = 0.3), and single/multiple patterns (χ: *p* = 0.8).

Because of the absence of significant differences in the “bad outcome” between high- and low-grade tumors, it was not possible to perform multilinear regression analyses to evaluate which factor could have an impact on the negative outcomes.

The multilinear regression, with which the effects of clinical parameters were analyzed, showed that in the HG group, all the studied variables (sex, age, margin, T, N, neck dissection, and adjuvant therapy) had a weak collective impact on the months of survival; however, the sample was small with very low power, and none of the variables revealed a statistically significant value.

The analysis of the same parameters in the low-grade group revealed overlapping results with the ones observed in cases of HG tumors, confirming that all these findings weakly affected (without statistically significant power) the months of survival.

The multilinear regression analysis performed on the histological characteristics of the tumor for evaluating the impact of these findings on the months of survival showed that in the HG group, all variables (size, necrosis, extraglandular, lymphovascular, atypical mitosis, neuroendocrine, and pleomorphism) had a moderate effect on the months of survival, without a statistically significant *p* value. The absence of a statistically significant value is related to the small sample of patients with HG tumors. 

In the low-grade group, the analysis of the same variables showed a weak collective effect on the months of survival, but even in this case, it was without a statistically significant *p* value.

## 4. Discussion

This study confirmed the predominance of AciCC among females, in full agreement with previously demonstrated findings. The age distribution (median age of 53.6 years) was concordant with the SEER data and with the findings of Gomez et al. and Hoffman et al. [19,20]. HG patients were older than those with low-grade neoplasms, as were those who experienced recurrence (mean, 72.2 years) compared with patients without it (mean, 52.28 years) (*p* = 0.0097).

The tumor size (pT parameter) and its extension inside the deep lobe of the parotid are universally considered relevant prognostic factors. We observed statistically significant differences in the tumor size between patients with recurrence (mean, 50.7 mm) and those without it (mean, 27.3 mm) (*p* = 0.0012). The same result was obtained comparing HG AciCC (mean, 42.6 mm) and low-grade AciCC (mean, 25.8 mm) (*p* = 0.000751), and this might confirm the pT prognostic impact.

Lymph nodes are rarely involved in the diffusion of the tumor; in fact, in our sample, only six patients (8%) showed lymph node metastases. In previous studies, van Weert et al. and Yibulayin et al. reported, respectively, a 9.2% and 2% rate of lymph node metastases [21,22]. In 82% of cases, ND was not performed because most of the patients did not have clinical lymph nodal involvement (cN+), in agreement with the current research indicating that performing this procedure is unnecessary in clinically N0 neck in the AciCC of the parotid gland. Occult neck metastasis in cN0 patients was identified in 1% of cases, supporting the concept of not routinely performing ND. Several controversies regarding lymph node involvement are reported in the literature. Grasl et al. [23] found positive lymph nodes in almost 15% of patients, while van Weert et al. did not obtain this finding at all [21]. Based on their results, Grasl et al. suggested considering elective neck dissection (END) even in cases with the absence of lymph nodes in the neck (N0).

In our study, low-grade AciCC patients presented lymph nodal involvement more than those with HG AciCC (*n* = 4, vs. *n* = 3), while patients with HG AciCC showed worse pN staging than those with low-grade AciCC (pN1; *n* = 0 vs. *n* = 3; pN2, a or b; *n* = 3 vs. *n* = 1). Based on these results, and the lack of multicast prospective analyses, routine END levels II-IV in patients with the AciCC of the parotid gland cannot be recommended.

We did not identify extra lymph node metastasis at the time of diagnosis, in full agreement with other researchers.

An adequate area free of disease, called free-of-disease surgical margins (i.e., the absence of cancer cells in the margin of the resected specimen), affects the survival time. In the present study, the margins of resection were wider in HG tumors than in low-grade tumors, with a statistically significant value (τ: *p* = 0.003). However, from a statistical point of view, there was no association between this parameter and the rate of recurrence and/or death. These results agree with those of Park YM et al. [24] but are in contrast to the findings of Gomez et al. [19] and Zenga et al. [25]. Notably, both Gomez and Zenga analyzed small cohorts of patients, comprising, respectively, 35 and 45 patients. Their difference in sample size from our sample could have impacted the results.

There is a lack of consensus about the use of adjuvant RT after AciCC surgery; empirically, adjuvant RT is suggested in case of (i) advanced AciCC, (ii) HG tumors, and (iii) the impossibility of obtaining free-of-disease margins during surgery.

In our study, adjuvant RT was used more in patients with HG AciCC than in those with low-grade tumors (τ: *p* = 0.02), without a significant impact on OS and DFS. The retrospective nature of our study did not allow us to investigate the reasons why some patients were referred to RT, while others were not.

We identified HG AciCC in 17% of cases, as in previous studies [21,26]; this prevalence was almost twice the one reported by larger case series [26,27].

We found an 8.7% rate of recurrence, which was lower than the one identified by Park et al. (18.6%), Kirschnik et al. (27.8%), and van Veert (15%); however, these case series were smaller than ours. The rate of distant metastases was low, both in low-grade tumors and HG AciCC, and the involvement of lateral skull base was quite rare, in full agreement with previous findings [28].

Several growth patterns have been described for the AciCC of the parotid gland; recently, Shah and Seethala [29] described a unique case of a squamoglandular variant of AciCC of the parotid gland, which had a similar blend of mucoepidermoid-like ad acinar elements and molecular phenotype of AciCC.

The solid pattern was the most common observed finding in this study, both in low-grade tumors (48%) and HG tumors (38%), followed by microcytic (14%) and papillary cystic (11%) patterns in HG AciCC, while trabecular (23%) and cribriform (23%) patterns followed the solid pattern in low-grade AciCC. No statistically significant differences were found among the two groups regarding the growth pattern (*p* = 0.33) nor number of tumors presenting multiple growth patterns (*p* = 0.8). The morphological pattern/s did not correlate with the grading, the recurrence rate, and the OS.

Some histological features have been historically correlated with aggressive behavior and the recurrence of AciCC. These include multiple mitotic figures, atypical cells, and stromal hyalinization. Comparing the histological results of the two groups, (as expected, and in accordance with previous reports), we found that HG tumors showed a statistically significant prevalence of specific histological features. In terms of defining a perfect “identikit” of HG patients (compared to the histological features of low-grade cases), the histologically relevant findings seem to be necrosis, extraglandular growth, LVI, atypical mitosis, and pleomorphism. The multilinear regression analysis of the histological characteristics of this type of cancer, which evaluated the impact of these findings on the months of survival, showed that in the HG group, all these variables (necrosis, extraglandular, lymphovascular, atypical mitosis, neuroendocrine, and pleomorphism) had a moderate effect on the months of survival, although without a statistically significant *p* value. However, the absence of statistically significant value could be related to the small sample of patients with HG tumors.

Recently, Xu et al. attempted to classify AciCC as low-grade, intermediate-grade, and HG tumors by using the mitotic index, necrosis, fibrosis at the frankly invasive front, and infiltrative border; they concluded that, while low- and intermediate-grade tumors behaved in a similar fashion, HG AciCC is characterized by a mitotic index of ≥5/10 HPFs and/or necrosis [15]. The results of our study confirmed the role of necrosis as a relevant indicator in the definition of Hg AciCC; in addition, mitotic index > 5 HPFs was observed more in HG AciCC (46%) than in low-grade tumors (2%). Nevertheless, these results were not consistent enough to consider this index in the definition of HG AciCC.

Our study is the first in which TIL expression is evaluated in a large cohort of AciCC tumors. TILs’ expression can be widely influenced by the tumor microenvironment (TME) and could have a role in the antitumoral or protumoral response. Recently, the prognostic meaning of TILs in head and neck cancers, especially head and neck squamous cell carcinoma (HNSCC), has been shown [30]. Some researchers suggested that the number and subtype of TILs expressed by tumor tissues could be useful to identify those patients eligible for immunotherapeutic approaches [31]. In major salivary gland tumors, the prognostic role of TILs is still debated. Recently, De Virgilio et al. sought to define the role of TILs and tumor-associated macrophages (TAMs) in salivary gland cancers; in this study, 80% of the tumors arose from the parotid gland, but only 8% (*n* = 2) of these tumors were AciCC. The major limitation of this study was the absence of data about the origin of the tumor (parotid or other salivary glands) [32]. According to their results, TILs can be associated with a higher likelihood of lymph node metastases.

We evaluated TILs in all the tissues, and no statistically significant differences were observed in the TIL values between low-grade and HG tumors (τ: *p* = 0.2), as previously reported [17]. Additionally, there was no correlation between TIL values and the presence of lymph node metastasis. Our results can support neither the harmful effect nor the protective role of TIL expression. Although our results are inconsistent regarding this finding, we believe that further (multicentric and larger) studies are necessary to correctly understand the role of TILs in major salivary gland tumors. Moreover, TILs should also be tested on staminal totipotent cells to understand their oncogenic potential (or lack thereof).

This study involves the largest European cohort of patients affected by AciCC of the parotid gland treated with surgery. We analyzed the outcomes prognostic factors of this rare tumor, and our results could be useful for clinical purposes to improve patient management. Furthermore, this study examined DSS; all follow-up data, with the longest follow-up regarding the AciCC of the parotid gland; and all the histological variables previously described.

### Limits of the Study

This study has some limitations. First, this work included data prior to 2010, when secretory carcinoma (SC), which exhibits a behavior clinically like AciCC, was still recognized as a distinct entity; nevertheless, all histological slides were reviewed by a senior coordinator pathologist to confirm the diagnosis of the AciCC of the parotid gland and to exclude SC via Pan-Trk immunostaining. Second, because of the retrospective nature of this study, the data about treatment decisions (surgery choice, management of the neck, and the selection of adjuvant therapy) were not available. Third, the cohort was quite small and unbalanced in terms of gender (women–men ratio = 3:1), and this could have impacted the results, which must be considered preliminary. Then, despite the agreement with previous research, the number of patients who experienced tumor recurrence was too low to set aside any statistical bias.

Another important limitation was the inability to use immunohistochemistry in NR4A3 (nuclear receptor subfamily 4 group A member 3), which is a fusion gene found in AciCC in 2019 [33], for studying our patients; unfortunately, not all centers had the technology to study this finding. For this reason, we did not include this analysis in our database, although we believe it has a prognostic value.

Finally, the two groups of patients stratified by histological grade (low grade and HG) were markedly disproportionate, which might have impacted the results of the statistical analyses; however, our breakdown percentage reflected the one already published.

## 5. Conclusions

This study was performed on the largest European cohort of patients affected by the AciCC of the parotid gland by examining DSS, including all follow-up information. Moreover, it has a long follow-up (probably the longest presented in the literature).

In most of the cases, AciCC tumors had an indolent behavior, with a low rate of cervical lymph node metastases, a small percentage of distant relapses, and optimal OS and DFS. With this multicenter, retrospective case series, perhaps one of the studies most representative of the AciCC of the parotid gland, we investigated the clinical–epidemiological–histological stratification of the patient at risk of poor outcomes. Based on the results of our study, we believe that the correct definition of HG AciCC should include the size of the tumor, necrosis, extraglandular spread, LVI, atypical mitosis, and cellular pleomorphism in addition to the high- and low-grade definitions.

Larger and international multicenter prospective studies are necessary to reach a unanimous consensus on the stratification risk of this histological entity.

## Figures and Tables

**Figure 1 cancers-15-05456-f001:**
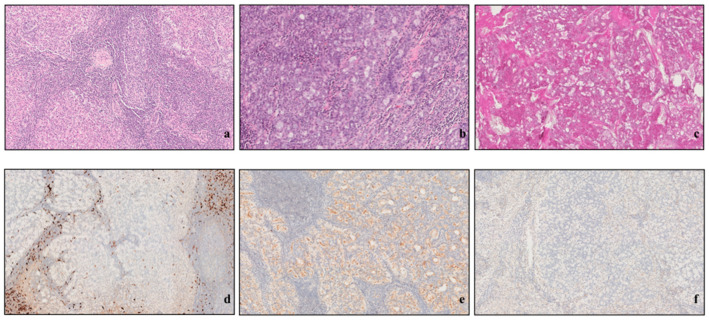
(**a**) EE2: tumor islands with intense stromal lymphocytic infiltration; (**b**) EE4: cytoplasmic basophilia of neoplastic cells; (**c**) PAS: cytoplasmatic positivity; (**d**) S100 negativity: poor positivity in lymphoid infiltrate (dendritic cells, antigen-presenting cells); (**e**) DOG1: discontinuous (but intense) membrane positivity; (**f**) p63 negativity.

**Figure 2 cancers-15-05456-f002:**
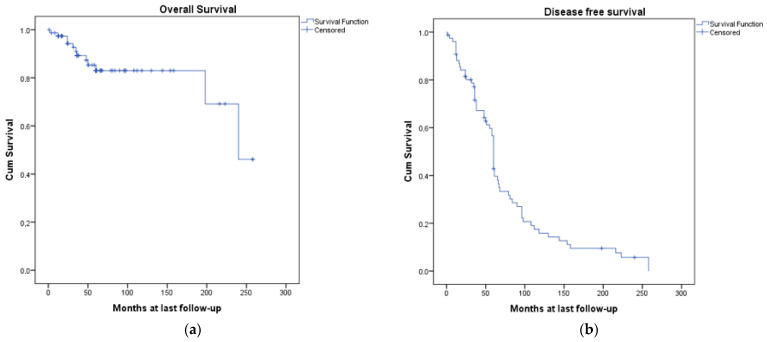
OS and DFS for the cohort of patients: (**a**) overall survival (OS) curve; (**b**) disease-free survival (DFS) curve.

**Table 1 cancers-15-05456-t001:** Characteristics of the cohort of patients with AciCC of the parotid gland included in the present work.

Variable		*n*	%
Participants		77	
Mean age (min–max; SD)	53.6 years (12–87–18.22)		
Gender	Female	53	71.4
	Male	24	28.6
Comorbidities	Hypertension	7	9
	Diabetes mellitus	5	6.5
	Asthma and/or pulmonary emphysema	4	5.2
	Neurological/mental disease	1	1.3
	Chronic renal disease	1	1.3
	Previous oncological disease (no head and neck region)	1	1.3
FNAB result	Not diagnostic Other parotid neoplasm Acinic cell carcinoma Not performed	40 16 13 8	58 23.2 18.8
Parotid Surgery (according to European Salivary Gland Society Classification)	I–IV (VII) (Total parotidectomy with facial nerve resection) I–IV (Total parotidectomy) I–II (Superficial parotidectomy) I–II–III (Superficial parotidectomy extended to inferior deep lobe)	1 33 39 4	5.3 37.8 50.6 5.3
Nodal dissection (ND)	No ND performed	64	82.1
	ND performed	13	16.9
	− Selective (II–IV)	8	61.5
	− Superselective (II or II–III)	3	23.1
	− mRND/RND (I–V)	2	15.4
Margin status	R0	34	44.1
	R1	22	28.6
	Rclose	21	27.3
Staging	Stage I	25	32.5
	Stage II	37	48
	Stage III	9	11.7
	Stage IVA	6	7.8
Adjuvant treatment	No	60	77.9
	Yes (RT)	17	22.1
Pathologic T classification (according to TNM classification—8th edition)	T1	26	33.8
	T2	40	51.9
	T3	8	10.4
	T4a	3	3.9
Pathologic N classification (according to TNM classification—8th edition)	N0	70	90.9
	N1	3	3.9
	N2a	2	2.6
	N2b	2	2.6
Recurrence	Local	6	
Mean follow-up (min–max; SD), months	67.4 (1–258; SD 59.39)		
Status at last follow-up	NED	62	80.5
	DOOC	8	10.4
	NED II	2	2.6
	DOD	3	3.9
	AWD	2	2.6

**Table 2 cancers-15-05456-t002:** Clinicopathological features of each patient who experienced recurrence.

Parameter	Patient 1	Patient 2	Patient 3	Patient 4	Patient 5	Patient 6
Age, gender	84, M	65, F	79, F	74, F	59, F	55, F
Type of surgery	I–IV	I–IV	I–IV	I–IV	I–II	I–IV
Ipsilateral ND	I–V	II–IV	None	None	None	II–IV
pTNM staging	pT4aN2aM0—IVa	pT3N2bM0—IVa	pT4aN0M0—IVa	pT2N0M0—II	pT1N0M0—I	pT1N1Mo—III
Resection margins	R0	R0	R1	R0	R1	Rclose
Adjuvant therapy	None	RT	CH	None	None	RT
Tumor diameter (mm)	90	45	60	35	59	15
Growth pattern/s	Trabecular + cribriform	Solid + cribriform	Solid + microcystic	Solid + microcystic	Cribriform	Microcystic
Grade	High grade	High grade	Low grade	Low grade	Low grade	Low grade
Necrosis	Microfocal (<1 mm)	Microfocal (<1 mm)	Absent	Macrofocal (>1 mm)	Absent	Absent
Perineural invasion	Present	Absent	Absent	Absent	Present	Absent
LVI	Focal (<2 figures)	Focal (<2 figures)	Absent	Absent	Absent	Absent
Extraglandular growth	Present	Present	Absent	Absent	Absent	Absent
Pleomorphism	Absent	Mild	Absent	Absent	Absent	Absent
Lymphoid stroma	Negative	Negative	Moderate (11–50%)	Moderate (11–50%)	Moderate (11–50%)	Absent
Atypical mitosis	Absent	Present	Absent	Absent	Absent	Absent
Mitotic index	2–4	2–4	0–1	>5	0–1	0–1
Neuroendocrine differentiation	Absent	Absent	Absent	Absent	Absent	Absent
Stromal hyalinization	Present	Present	Present	Present	Present	Absent
TILs (%)	5%	0%	0%	30%	20%	1
Significant ki67 (>15%)	20%	DNA	DNA	30%	DNA	DNA
Number of recurrences, sites, and number of months after initial diagnosis	1; parotid area and external auditory canal; 4 months	1; parotid area; 5 months	1; lateral skull base; 1 month	2; a. parotid area, external auditory canal, tympanic cavity, mastoid, sigmoid sinus, pontocerebellar angle, 11 months; b. site of previous left petrosectomy, foramen lacerum, foramen magnum, 25 months	1; parotid area, 11 months	1; parotid area, 26 months
Status at last follow-up	16 months, NEDII	35, AWD	2, DOD	31, DOD	38, NEDII	60, DOD

**Table 3 cancers-15-05456-t003:** Pathological features of the tumors’ slides included in the study.

Parameter	Categories	Number of Patients (*n*)
Tumor diameter	0–1.9 cm	36
	2–4.9 cm	32
	5–10 cm	9
Growth pattern	Solid	36
	Trabecular	4
	Cribriform	8
	Microcystic	12
	Papillary cystic	8
	Follicular	9
	Multiple	39
Grade	Low	64
	High	13
Resection margins	R0	32
	Rclose	22
	R1	23
Necrosis	Absent	54
	Microfocal < 1 mm	7
	Macrofocal > 1 mm	6
	Diffuse	10
Perineural invasion	Absent	64
	Present	13
Lymphovascular invasion	Absent	56
	Focal (<2 figures of LVI)	14
	Diffuse (>2 figures of LVI)	7
Extraglandular growth	Absent	49
	Present	28
Pleomorphism	Absent	46
	Mild	11
	Moderate	10
	Severe	10
Lymphoid stroma	Negative (<1%)	34
	Mild (1–10%)	27
	Moderate (11–50%)	16
	Severe (>50%)	0
Atypical mitosis	Absent	67
	Present	10
Mitotic Index (per 10HPFs)	0–1	50
	2–4	20
	>5	7
Neuroendocrine differentiation	Absent	75
	Present	2
Stromal Hyalinization	Absent	47
	Present	30
TILs (%)	0	20
	1–4	14
	5–9	12
	10–19	18
	>20	13
Ki67 (%)	0–4	5
	5–9	7
	>10	13
	DNA	52

**Table 4 cancers-15-05456-t004:** Summary of the comparison between the group of patients who experienced recurrence and those without relapse.

Parameter	Group A (AciCC Patients Who Experienced Recurrence)	Group B (AciCC Patients with no Recurrence)	Two-Tailed *p* Value
Size of the cohort (n)	6	71	
Age (mean, range, SD), yr	72.2	52.28	0.0097
	(55–84; SD 9.1)	(72–87; SD 18.11)	
Gender	5 F; 1 M	48 F; 23 M	
Type of surgery			
I–II	1	38	
I–IV	5	24	
Ipsilateral ND			
No	3	61	
Yes	3	10	
Pathologic T, *n*			
pT1	2	24	
pT2	1	39	
pT3	1	7	
pT4a	2	1	
Pathologic N, *n*			
pN0	3	67	
pN1	1	2	
pN2(a or b)	2	2	
Staging			
I	1	24	
II	1	36	
III	1	8	
IVa	3	3	
Resection margins			
R0	3	27	
Rclose	1	30	
R1	2	22	
Adjuvant RT/CH			
No	3	55	
Yes	3	16	
Tumor diameter (range, mean), mm	50.7	27.3	0.0012
	(15–90; SD 23.28)	(4–73; SD 15.7)	
Main growth pattern, *n*			
Solid	3	36	
Trabecular	1	4	
Cribriform	1	7	
Microcystic	1	12	
Papillary cystic	0	3	
Follicular	0	9	
Multiple growth patterns, *n*	4	35	
Grade, *n*			
Low grade	4	60	
High grade	2	11	
Necrosis, *n*			
Absent	4	51	
Present	2	20	
Perineural invasion, *n*			
Absent	4	60	
Present	2	11	
LVI, *n*			
Absent	4	52	
Present	2	19	
Extraglandular growth, *n*			
Absent	4	44	
Present	2	27	
Pleomorphism, *n*			
Absent	5	43	
Present	1	29	
Lymphoid stroma, *n*			
Absent	5	31	
Present	1	40	
Atypical mitosis, *n*			
Absent	5	62	
Present	1	9	
Mitotic index, *n*			
0–1	3	45	
2–4	2	21	
>5	1	8	
Stromal hyalinization, *n* Absent Present	1	45	
	5	26	
TILs (%), *n*			
0	2	18	
1–4	1	12	
5–9	1	23	
10–19	0	6	
>20	2	12	
Follow-up (range, mean, SD), months	30.3	67.4	0.1338
	(2–60; SD 18.11)	(1–258; SD 59.39)	

**Table 5 cancers-15-05456-t005:** Summary of the comparison between HG AciCC and low-grade tumors.

Parameter	Group A (Low-Grade AciCC)	Group B (High-Grade AciCC)	Two-Tailed *p* Value
Size of the cohort (*n*)	64 (83%)	13 (17%)	
Age (mean, range, SD), yr	49.93 (12–87; SD 17.28)	71.7 (55–84; SD 10.11)	<0.0001
Gender	45 F (70.3%); 19 M (29.7%)	8 F (61.5%); 5 M (38.5%)	0.5
Comorbidities	4	3	
Hypertension	3	2	
Diabetes mellitus	2	2	
Asthma and/or pulmonary emphysema			
Neurological/mental disease	1	0	
Chronic renal disease	1	0	
Previous oncological disease (not in the head and neck region)	1	0	
Type of surgery			0.42
I–II	35 (54.7%)	4 (31%)	
I–IV	23 (35.9%)	6 (46%)	
Ipsilateral ND			0.00182
No	57 (89%)	7 (54%)	
Yes	7 (11%)	6 (46%)	
Pathologic T, *n*			0.000093
pT1	26 (41%)	0	
pT2	32 (50%)	8 (61%)	
pT3	5 (8%)	3 (23%)	
pT4a	1 (1%)	2 (16%)	
Pathologic N, *n*			0.00119
pN0	60 (94%)	10 (77%)	
pN1	3 (5%)	0	
pN2(a or b)	1 (1%)	3 (23%)	
Staging			
I	25 (39%)	0	
II	29 (45%)	8 (61.5%)	
III	8 (12.5%)	1 (8%)	
IVa	2 (3.5%)	4 (30.5%)	
Resection margins			0.003
R0	29 (45%)	3 (23%)	
Rclose	20 (31%)	1 (8%)	
R1	15 (24%)	9 (69%)	
Adjuvant RT/CH			0.0206
No	50 (78%)	7 (54%)	
Yes	14 (22%)	6 (46%)	
Tumor diameter (range, mean), mm	25.8 (4–70; SD 15.5)	42.6 (18–90; SD 20.54)	0.000751
Main growth pattern, *n*			0.33
Solid	31 (48%)	5 (38%)	
Trabecular	3 (5%)	1 (8%)	
Cribriform	5 (8%)	3 (23%)	
Microcystic	9 (14%)	3 (23%)	
Papillary cystic	7 (11%)	1 (8%)	
Follicular	9 (14%)	0	
Multiple growth patterns, *n*	32 (50%)	7 (54%)	0.8
Necrosis, *n*			0.01
Absent	51 (80%)	3 (23%)	
Present	13 (20%)	10 (77%)	
Perineural invasion, *n*			0.1426
Absent	55 (86%)	9 (54%)	
Present	9 (14%)	4 (46%)	
LVI, *n*			0.006
Absent	51 (80%)	5 (38%)	
Present	13 (20%)	8 (62%)	
Extraglandular growth, *n*			0.00997
Absent	44 (69%)	4 (46%)	
Present	20 (31%)	9 (54%)	
Pleomorphism, *n*			0.00334
Absent	49 (77%)	3 (23%)	
Present	21 (23%)	10 (77%)	
Lymphoid stroma, *n*			0.12
Absent	27 (42%)	7 (54%)	
Present	37 (58%)	6 (46%)	
Atypical mitosis, *n*			<0.00001
Absent	61 (95%)	6 (46%)	
Present	3 (5%)	7 (54%)	
Mitotic index, *n*			0.33
0–1	48 (75%)	2 (16%)	
2–4	15 (23%)	5 (38%)	
>5	1 (2%)	6 (46%)	
Stromal hyalinization, *n*			0.6345
Absent	39 (61%)	7 (54%)	
Present	25 (39%)	6 (46%)	
TILs (%), *n*			0.2619
0	17 (26%)	3 (23%)	
1–4	14 (22%)	0	
5–9	7 (11%)	5 (38%)	
10–19	13 (20%)	2 (16%)	
>20	13 (20%)	3 (23%)	
Follow-up (range, mean, SD), months	71.9 (1–258; SD 63.13)	45 (12–112; SD 25.83)	0.06
Status at the last follow-up, *n*			
NED	54 (84%)	9 (69%)	
NEDII	1 (2%)	1 (8%)	
AWD	0	1 (8%)	
DOD	3 (5%)	0	
DOOC	6 (9%)	2 (15%)	

## Data Availability

Data are contained within the article.

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
