# Peer review of "A Retrospective Multicenter Italian Analysis of Epidemiological, Clinical and Histopathological Features in a Sample of Patients with Acinic Cell Carcinoma of the Parotid Gland"

_cancers, 2023, doi:10.3390/cancers15225456_

Round 1

Reviewer 1 Report (Previous Reviewer 2)

Comments and Suggestions for Authors

I received an article for review that was resubmitted once again. I see that the corrections indicated by the reviewers, including my corrections, were accepted by the authors of the text. The necessary corrections have been incorporated into the text. In such a situation, I see no other solution than to congratulate the authors on their interesting and innovative work and give the green light for the editors to publish this manuscript.

Author Response

Dear Reviewers,

Thanks for reviewing our manuscript and for the valuable comments that helped us clarify some relevant aspects that were missed or unclear in the first version of the paper.

We have read carefully your comments and made the changes to address comments and concerns.

We hope that the changes made in the revised manuscript and responses provided below have adequately addressed the reviewers’ comments and made this paper stronger.

Reviewer 1

  • “I received an article for review that was resubmitted once again. I see that the corrections indicated by the reviewers, including my corrections, were accepted by the authors of the text. The necessary corrections have been incorporated into the text. In such a situation, I see no other solution than to congratulate the authors on their interesting and innovative work and give the green light for the editors to publish this manuscript”.

Thank for your valuable comments on the first review, which helped to improve this work.

Reviewer 2 Report (New Reviewer)

Comments and Suggestions for Authors

This study is a comprehensive clinical and pathological review of 77 acinic cell carcinomas of the parotid gland.

This provides valuable insight into the clinical and pathological aspects of this rare type of parotid cancer.

I recommend just one thing to be revised.

Pathological terms high-grade and low-grade tumor should be clearly separated from the clinical terms high-risk and low-risk tumors, in abstract and introduction.

I suggest a different term for clinical outcomes, rather than high-grade or low-grade tumors.

For example, high recurrence risk or low recurrence risk, or high-risk or low-risk.

[END]

Comments on the Quality of English Language

Minor editing of English language required

Author Response

Dear Reviewers,

Thanks for reviewing our manuscript and for the valuable comments that helped us clarify some relevant aspects that were missed or unclear in the first version of the paper.

We have read carefully your comments and made the changes to address comments and concerns.

We hope that the changes made in the revised manuscript and responses provided below have adequately addressed the reviewers’ comments and made this paper stronger.

Reviewer 2

  • “This study is a comprehensive clinical and pathological review of 77 acinic cell carcinomas of the parotid gland. This provides valuable insight into the clinical and pathological aspects of this rare type of parotid cancer. I recommend just one thing to be revised. Pathological terms high-grade and low-grade tumor should be clearly separated from the clinical terms high-risk and low-risk tumors, in abstract and introduction. I suggest a different term for clinical outcomes, rather than high-grade or low-grade tumors. For example, high recurrence risk or low recurrence risk, or high-risk or low-risk”.

Thank for your valuable comment; we have changed the terms in accordance with your suggestions in the abstract and in the introduction.

English has been carefully corrected throughout the text.

This manuscript is a resubmission of an earlier submission. The following is a list of the peer review reports and author responses from that submission.

Round 1

Reviewer 1 Report

Comments and Suggestions for Authors

The authors concluded that the correct definition of HG AciCC should include the size of the tumor, necrosis, extraglandular spread, LVI, atypical mitosis, and cellular pleomorphism. However, iIn our view, tumor grading is a histologic definition that describes how abnormal the cancer cells are under the microscope and that parameters such as tumor size, LVI   and extrglandular spread should be considered as risk factors rather than grading parameters.

In summary, I think that this study requires significant revision and reevaluation to ensure a more rigorous and accurate assessment of AciCC and its grading criteria.

The authors of this study examined a relatively large cohort of AciCC and conducted an analysis on its prognostic factors. Their primary focus was on grading AciCC but there was a lack of clarity on the specific grading system. Of note they used a grading established in a 2009 by Skalova et al, which documented 9 AciCCs with HGT. However, it worth mentioning that the terminology “HG” was used interchangeably with HGT, although there wasn’t a consistent and clear set of criteria to differentiated HG tumors and those with HGT.

A more recent study conducted by Xu et al, which examined s substantial sample pf 117 AciCCs, defined HG based on parameters such as mitotic index and tumor necrosis. Interestingly, the authors of the current study did not discuss this more recent study and classify tumors with necrosis or high mitotic count as HG. In our perspective, categorizing salivary gland tumors with necrosis as low grade appears to be incompatible. With the current understanding

One noteworthy concern is whether a statistician was involved in this study, as the conclusion and results were based on arbitrary tumor grade stratification based and not grounded prognostic events or survival data. 

Author Response

Dear Reviewer,

Thanks for reviewing our manuscript and for the valuable comments that helped us clarify some relevant aspects that were missed or unclear in the first version of the paper. 

We have carefully read your comments and made the changes to address comments and concerns. 

We hope that the changes made in the revised manuscript and responses in bolded Italic provided below have adequately addressed the reviewer’s comments and made this paper stronger.

  1. The authors concluded that the correct definition of HG AciCC should include the size of the tumor, necrosis, extraglandular spread, LVI, atypical mitosis, and cellular pleomorphism. However, in our view, tumor grading is a histologic definition that describes how abnormal the cancer cells are under the microscope and that parameters such as tumor size, LVI and extrglandular spread should be considered as risk factors rather than grading parameters. In summary, I think that this study requires significant revision and reevaluation to ensure a more rigorous and accurate assessment of AciCC and its grading criteria.
    The authors of this study examined a relatively large cohort of AciCC and conducted an analysis on its prognoti'c factors. Their primary focus was on grading AciCC but there was a lack of clarity on the specific grading system. Of note they used a grading established in a 2009 by Skalova et al, which documented 9 AciCCs with HGT. However, it worth mentioning that the terminology “HG” was used interchangeably with HGT, although there wasn’t a consistent and clear set of criteria to differentiated HG tumors and those with HGT. 

Thanks for your comment. This was clearly a mistake. We have made the changes throughout the manuscript.

  1. A more recent study conducted by Xu et al, which examined s substantial sample pf 117 AciCCs, defined HG based on parameters such as mitotic index and tumor necrosis. Interestingly, the authors of the current study did not discuss this more recent study and classify tumors with necrosis or high mitotic count as HG. In our perspective, categorizing salivary gland tumors with necrosis as low grade appears to be incompatible.

Thanks for your valuable comment. We strongly believe that NR4A3 (Nuclear Receptor Subfamily 4 Group A member 3) immunohistochemistry could represent a useful (diagnostic and prognostic) tool, and we added this limitation of the study in the proper paragraph.

3. One noteworthy concern is whether a statistician was involved in this study, as the conclusion and results were based on arbitrary tumor grade stratification based and not grounded prognostic events or survival data.

Thanks for your comment. One of the authors (Arianna Di Stadio) who was involved in the study design and the analyses of the data, earned a Master in Biostatistics at the University of Harvard. Because of her expertise in this field, she substituted the statistician. For more info you can reach her at arianna.distadio@unict.it.

Reviewer 2 Report

Comments and Suggestions for Authors

The article touches on a very important and interesting subject of the very rare parotid tumor – acinic cell carcinoma. I think that the manuscript has been prepared very well in terms of chapter contents and collected data. It is very informative and easy to read.

The paper presented to me for review is a multicenter retrospective study on a very rare parotid tumor, which is acinic cell carcinoma. The work has a typical layout. The authors examined a group of 77 patients, which, considering the fact that this is a very rare tumor, is a large group. The authors presented the results of their research in a tabular version, which makes it much easier for the reader to analyze the collected data. Particularly noteworthy is the very precise and extensive section on the anatomopathological aspects of this tumor. The details of the implemented surgical procedures are described well. Aspects concerning the occurrence of other systemic diseases (comorbidities) in the patients included in the study were not sufficiently presented. Considering that even in the title of the work, the authors indicate that the work concerns the analysis of clinicopathological factors, this issue needs to be supplemented. After such an assessment, it may also be possible to find some correlations between the occurrence of this tumor and other conditions. Such observations would be very valuable.

As for the linguistic aspect of the paper – it is generally good but requires some editorial work, preferably by a professional translator/editor or a native speaker, because some parts of the text contain unnatural grammatical structures or a peculiar wording.

I hope to see this manuscript’s updated version with proper adjustments made according to my suggestions.

Comments on the Quality of English Language

 As for the linguistic aspect of the paper – it is generally good but requires some editorial work, preferably by a professional translator/editor or a native speaker, because some parts of the text contain unnatural grammatical structures or a peculiar wording.

Author Response

Dear Reviewer,

Thanks for reviewing our manuscript and for the valuable comments that helped us clarify some relevant aspects that were missed or unclear in the first version of the paper. 

We have carefully read your comments and made the changes to address comments and concerns. 

We hope that the changes made in the revised manuscript and responses in bolded Italic provided below have adequately addressed the reviewer’s comments and made this paper stronger.

  1. The article touches on a very important and interesting subject of the very rare parotid tumor – acinic cell carcinoma. I think that the manuscript has been prepared very well in terms of chapter contents and collected data. It is very informative and easy to read.
    The paper presented to me for review is a multicenter retrospective study on a very rare parotid tumor, which is acinic cell carcinoma. The work has a typical layout. The authors examined a group of 77 patients, which, considering the fact that this is a very rare tumor, is a large group. The authors presented the results of their research in a tabular version, which makes it much easier for the reader to analyze the collected data. Particularly noteworthy is the very precise and extensive section on the anatomopathological aspects of this tumor. The details of the implemented surgical procedures are described well. Aspects concerning the occurrence of other systemic diseases (comorbidities) in the patients included in the study were not sufficiently presented. Considering that even in the title of the work, the authors indicate that the work concerns the analysis of clinicopathological factors, this issue needs to be supplemented. After such an assessment, it may also be possible to find some correlations between the occurrence of this tumor and other conditions. Such observations would be very valuable. 

Thanks for your detailed comments. In the Table 1, we added all the info about main comorbidities of the patients included in the study. Although these data are available for all patients, we did not believe that a statistical analysis on these comorbidities/ factors could be useful. The reasons of this belief are as first it would be necessary to analyze a series of other markers (i.e pro-inflammatory citokines or miR) to determine the cause-effect relationship between i.e hypertension and recurrence -in case of identification of a relationship between this comorbidity and the recurrence; without these additional investigations, in case of a (hypothetical) correlation between hypertension and recurrence the explanation of this relationship could only be speculative. As additional, adding a series of general comorbidities without a clear rationale could only be confusing for the readers.  Finally, in our sample the comorbities were equally distributed between HG-AciCC and low-grade AciCC. Because we understand that our title might be confusing regarding this point, we changed it as follow “Acinic Carcinoma of parotid gland: more than a simple high versus low grade classification. A multicenter study of a rare tumor.” 

2. As for the linguistic aspect of the paper – it is generally good but requires some editorial work, preferably by a professional translator/editor or a na've speaker, because some parts of the text contain unnatural grammatical structures or a peculiar wording. I hope to see this manuscript’s updated version with proper adjustments made according to my suggestions. 

Thanks for your comment. The English has been revised by a professional expert in medicine. 

Round 2

Reviewer 1 Report

Comments and Suggestions for Authors

Dear Authors, Thank you for submitting your manuscript "High grade versus low grade is still a valid concept in acinic cell carcinoma of parotid gland? Results from a multicenter clinicopathological Italian analysis" to Cancers. We appreciate the time you invested in preparing the submission and revising the manuscript. After careful review, we regret to inform you that we are unable to accept your manuscript for publication. We encourage you to consider addressing the mentioned concerns. Thank you for your work and wish you the best. 

Reviewer 2 Report

Comments and Suggestions for Authors

I am very pleased to see the authors’ approach and attitude towards improving the manuscript. All of my feedback has been adressed and appropriate adjustments have been implemented into the text. The editorial work was extensive and benefited the article so that I can fully accept it in the current form and submit for final approval to be published. Best regards.